# Severe Lymphatic Disorder and Multifocal Atrial Tachycardia Treated with Trametinib in a Patient with Noonan Syndrome and SOS1 Mutation

**DOI:** 10.3390/genes13091503

**Published:** 2022-08-23

**Authors:** Michele Lioncino, Adelaide Fusco, Emanuele Monda, Diego Colonna, Michelina Sibilio, Martina Caiazza, Daniela Magri, Angela Carla Borrelli, Barbara D’Onofrio, Maria Luisa Mazzella, Rossella Colantuono, Maria Rosaria Arienzo, Berardo Sarubbi, Maria Giovanna Russo, Giovanni Chello, Giuseppe Limongelli

**Affiliations:** 1Inherited and Rare Cardiovascular Diseases Unit, Department of Translational Medical Sciences, University of Campania “Luigi Vanvitelli”, Monaldi Hospital, 81031 Naples, Italy; 2Adult Congenital Heart Disease Unit, Monaldi Hospital, 81031 Naples, Italy; 3Department of Pediatrics, Santobono-Pausilipon Children’s Hospital, 81031 Naples, Italy; 4Department of Neonatal Intensive Care, Monaldi Hospital, 81031 Naples, Italy; 5Paediatric Cardiology Unit, “L.Vanvitelli” University—Monaldi Hospital, 81031 Naples, Italy; 6Institute of Cardiovascular Sciences, University College of London and St. Bartholomew’s Hospital, London WC1E 6DD, UK

**Keywords:** Noonan syndrome, lymphangectasia, hypertrophic cardiomyopathy, RASopathies, MEK1, trametinib

## Abstract

Noonan syndrome (NS) is a multisystemic disorder caused by germline mutations in the Ras/MAPK cascade, causing a broad spectrum of phenotypical abnormalities, including abnormal facies, developmental delay, bleeding diathesis, congenital heart disease (mainly pulmonary stenosis and hypertrophic cardiomyopathy), lymphatic disorders, and uro-genital abnormalities. Multifocal atrial tachycardia has been associated with NS, where it may occur independently of hypertrophic cardiomyopathy. Trametinib, a highly selective MEK1/2 inhibitor currently approved for the treatment of cancer, has been shown to reverse left ventricular hypertrophy in two RIT1-mutated newborns with NS and severe hypertrophic cardiomyopathy. Severe lymphatic abnormalities may contribute to decreased pulmonary compliance in NS, and pulmonary lymphangiectasias should be included in the differential diagnosis of a newborn requiring prolonged oxygen administration. Herein we report the case of a pre-term newborn who was admitted to our unit for the occurrence of severe respiratory distress and subentrant MAT treated with trametinib.

## 1. Introduction

Noonan syndrome (NS) is a multisystemic disorder caused by germline mutations in the Ras/MAPK cascade, causing a broad spectrum of phenotypical abnormalities, including abnormal facies, developmental delay, bleeding diathesis, congenital heart disease (mainly pulmonary stenosis and hypertrophic cardiomyopathy), lymphatic disorders, and uro-genital abnormalities [1,2]. With an estimated prevalence of 1:2500 live births, NS is the most common RASopathy and the second most common cause of CHD after trisomy 21 [1]. Patients with NS show typical facial features such as hypertelorism, a wide forehead, down-slanting palpebral fissures and low-set ears. Growth retardation is common and requires hormone treatment in a significant number of cases [2]. Cryptorchidism, dermatological abnormalities and a predisposition to several malignancies, such as juvenile myelomonocytic leukemia, embryonal rhabdomyosarcoma or acute myelogenous leukemia, have been reported [3]. The first pathogenic gene associated with NS was identified in 2001 as the tyrosine phosphatase PTPN11, which resulted in the hyper-activation of the RAS/MAPK cascade. Guanyl nucleotide exchange factor SOS1 mutations are the second most prevalent cause of NS, and taken collectively, mutations in these two genes account for almost 65% of cases [4,5]. Currently, no specific target treatment is available for NS.

Multifocal atrial tachycardia has been associated with NS, where it may occur independently of hypertrophic cardiomyopathy [6]. Trametinib, a highly selective MEK1/2 inhibitor currently approved for the treatment of cancer, has been shown to reverse left ventricular hypertrophy in two RIT1-mutated newborns with NS and severe hypertrophic cardiomyopathy [7], and has recently been adopted in the treatment of NS-associated multifocal atrial tachycardia (MAT) [8]. Severe lymphatic abnormalities may contribute to decreased pulmonary compliance in NS, and pulmonary lymphangiectasias should be included in the differential diagnosis of a newborn requiring prolonged oxygen administration. Herein we report the case of a pre-term newborn who was admitted to our unit for the occurrence of severe respiratory distress and subentrant MAT treated with trametinib.

## 2. Case Description

A pre-term newborn (gestational age: 34 weeks) was admitted to our Pediatric Intensive Care Unit with the diagnosis of acute respiratory distress and NS.

During pregnancy, the identification of polyhydramnios and bilateral pleural effusion triggered genetic testing, which showed a heterozygous missense mutation in SOS1 gene (c.1655G>C, p.Arg552Thr, NM_005633.4), classified as pathogenic for NS (class V according to American College of Medical Genetics and Genomics (ACMG criteria)). Parental NGS testing revealed that the mutation arose de novo.

After urgent caesarean delivery the patient was assisted with positive expiratory pressure ventilation (fraction of inspired oxygen 30%) and then intubated and sedated. Because of bilateral pleural effusion, a thoracic drainage was positioned, and surfactant was administered on the second day of life. Laboratory examination of pleural fluid obtained by perinatal thoracentesis demonstrated a milky appearance and high lymphocyte and triglyceride concentrations (>110 mg/dL), supporting the diagnosis of chylothorax.

Electrocardiographic monitoring showed incessant multifocal atrial tachycardia with aberrant atrioventricular conduction, and signs of acute heart failure were identified.

The patient was moved to our paediatric intensive care unit, which represented the hub for inherited and rare cardiovascular diseases in infancy and childhood.

At admission, he was apyretic, apparently normoperfused and with stable vital parameters. His heart rate was 175 bpm, blood pressure was 74/35 mmHg, and respiratory rate was 45/minutes. He was supported with mechanical ventilation (fraction of inspired oxygen: 0.3), and his partial arterial oxygen pressure was 77 mmHg at blood gas analysis.

Physical examination showed abnormal facies, cryptorchidism, lower-set ears, a broad neck, teletelia, brachydactyly, and deeply grooved palm furrows, consistent with the diagnosis of NS. Echocardiogram showed two small muscular interventricular septal defects with mild left-to-right shunt and mild pulmonary stenosis with dysplastic pulmonary valve. Diastolic function and cardiac wall thickness were within the reference limits. Because of multifocal atrial tachycardia with rapid ventricular response, propranolol (1 mg/kg t.i.d.) was introduced and serial 24 h ECG Holter monitoring was indicated. For the persistence of subentrant episodes of MAT, flecainide (5 mg/kg b.i.d.) and then intravenous amiodarone (5 mcg/kg minute) were added on top of β-blocker therapy (Figure 1). Despite anti-arrhythmic therapy, thyroid function tests remained within the reference ranges.

Ultrasound lung scan performed after admission detected significant foci of lung consolidation with air bronchogram, involving basal fields bilaterally and medium fields of the right lung, associated with right pleural effusion. Pulmonary high-resolution CT scan (HRCT) showed bilateral parenchymal lung consolidation, involving both superior and inferior lobes, mainly with posterior extension, associated with diffuse *ground glass* opacities and smooth interlobular septal thickening, consistent with pulmonary lymphangiectasia/acute respiratory distress (Figure 2A).

Blood cultures and serum antibodies excluded active infection, and C-reactive protein remained negative during hospital stay. Because the patient showed a persistent need for mechanical ventilation and severe bilateral involvement was detected on chest HRCT, treatment with dexamethasone was started, with partial recovery of respiratory parameters. Although the patient was extubated on day 33 and continuous positive airway pressure (CPAP) was initiated, a progressive decline in respiratory function caused re-intubation on day 56. In the suspicion of NS-associated lymphangiectasia, lymphoscintigraphy was proposed for diagnostic confirmation. Nevertheless, the patient’s parents did not provide their informed consent.

Because of sub-entrant, drug-resistant episodes of MAT and progressive decline in respiratory function, management options were discussed by a multidisciplinary heart team, and the decision was to pursue compassionate administration of pharmacological MEK inhibitors. After approval by Institutional Ethics Committee of our institution (AORN dei Colli-Monaldi Hospital, Naples, Italy, project identification code AOC 0015028-2021), trametinib, a selective inhibitor of MEK1/2, was obtained by the Novartis Management Program. The study was conducted according to the principles of the declaration of Helsinki.

Treatment with trametinib (0.02 mg/kg daily) was started on week 9, after written informed consent was provided by the parents. A screening protocol, including ophthalmological, cardiological, dermatological, radiographic and haematological evaluation, was performed to monitor for possible side effects based on clinical trials (Appendix A). After 4 days of treatment, lung ultrasound examination showed the absence of pleural effusion. Lung B-line artifacts were detected in medium and basal pulmonary fields bilaterally, although mostly involving the right lung. The patient did not show any further episode of MAT within 72 h from treatment initiation.

A significant improvement in respiratory parameters was recorded, and the patient was extubated on week 10 and CPAP ventilation started, followed by O_2_ administration by nasal cannula until week 12, when the patient was left on room air. Follow-up chest CT scan showed a significant reduction in lung consolidation areas (Figure 2B). After 4 months follow-up, the patient was in good clinical condition and no further episodes of MAT have been recorded. Our patient did not develop hypertrophic cardiomyopathy during follow-up echocardiographic evaluations. Anti-arrhythmic therapy has been withdrawn and no side effect known to be associated with trametinib has been recorded. Because of the occurrence of feeding difficulties and laryngomalacia consequent to prolonged endotracheal intubation, percutaneous endoscopic gastrostomy was performed.

## 3. Discussion

Molecular target therapy represents a promising modality for the treatment of inherited and rare cardiovascular disease. Due to the extensive knowledge about the possible pathogenic mechanisms, RASopathies may represent a paradigm for the use of target, tailored molecules [7,9]. Severe lymphatic abnormalities have been shown to be common among patients with NS (up to 15%) and may carry significant prognostic implications for the complexity and poor availability of specific therapies. Biko et al. demonstrated that patients with NS may exhibit central lymphatic anomalies, such as the absence or duplication of the thoracic duct, retrograde intercostal flow and pulmonary lymphatic perfusion [10]. Although it is unclear if certain genotypes could confer an higher risk of lymphatic abnormalities among patients with NS, *SOS1* missense mutations have been associated with pulmonary and gastrointestinal lymphangectasias [9]. Interestingly, the administration of trametinib was associated with a rapid remodelling of the entire central lymphatic system.

In our case, the presence of lymphatic abnormalities was not demonstrated by means of MR-lymphangiography or lymphoscintigraphy. Nevertheless, CT scan showed diffuse ground glass opacities and smooth interlobular septal thickening, which have been associated with pulmonary lymphangiectasias [11]. Although the occurrence of bronchopulmonary dysplasia (BD) could not be excluded in our patient, a rapid improvement in respiratory parameters after the administration of trametinib seems unlikely in BD.

Our patient did not develop any evidence of hypertrophic cardiomyopathy during serial follow-up echocardiographic evaluations. This is of great clinical interest, as the regression of significant diastolic dysfunction resulting in pulmonary oedema may represent an important alternative hypothesis to consider for improvement in pulmonary pathology with the initiation of trametinib rather than lymphatic-mediated effects.

Our patient presented incessant episodes of MAT, which were treated with the combination of sotalol, flecainide and amiodarone. MAT has been frequently described among patients with RASopathies, where it occurs independently of HCM phenotype and shows high rates of recurrence with medical therapy alone [6]. Propranolol and digoxin are not useful to prevent MAT, whereas the use of flecainide or amiodarone has been shown to be more effective in sinus rhythm restoration and maintenance. Trametinib has been shown to rapidly ameliorate the recurrence rate of MAT in patients with *Raf1* mutations and triple antiarrhythmic therapy with diltiazem, flecainide and amiodarone [8]. However, the mechanisms of its efficacy are not completely understood. In our case, no further episode of MAT was recorded within 72 h from treatment initiation. As trametinib reaches steady-state concentrations within 48–72 h, we hypothesized that higher plasma concentrations are required to exert its antiarrhythmic effect. The temporal association between trametinib and MAT resolution also strengthens the hypothesis of a direct MEK1 pathway rather than an improvement in secondary respiratory or other conditions affecting the patient’s clinical state.

Different pathway mechanisms may underlie the association with MAT and NS. Previous studies suggested that MAT results from dysregulated calcium handling, partly mediated by inappropriate calcium release by ryanodine receptor (Ryr2), causing delayed after-depolarization and increased triggered activity [12,13,14]. Therefore, the use of flecainide, a sodium channel blocker, may exert its efficacy both directly, by reducing calcium release from the Ryr2 receptor, and indirectly, by inhibiting Na^+^ influx through voltage-gated sodium channels (I_Na_) and subsequently increasing calcium efflux via a sarcolemmal Na^+^/Ca^2+^ exchange protein (NCX) [15]. Animal models have shown that Ryr2 is hyperphosphorylated at basal levels; nevertheless, in myocytes from hearts with atrial arrhythmias, calcium-calmodulin-dependent protein kinase II and protein kinase A may mediate hyperactivation and may contribute to progressive myocardial dysfunction [16,17,18,19]. Murine models of atrial fibrillation have demonstrated a cross talk between PKC and MAPK pathways in cardiac fibroblasts. In particular, the treatment of cell cultures with a PKC activator, PMA, was associated with a global increase in protein synthesis and ERK expression in fibroblasts from AF rats but not from healthy controls. Interestingly, the use of the ERK inhibitor trametinib attenuated protein synthesis and PKC activation both in healthy and AF rats [18]. In our patient, the biochemical behaviour of the Arg455 mutation suggested that the maintenance of SOS1 in its autoinhibited conformation was impaired, leading to increased flow through the Ras/MAPK pathway [19].

In our case, trametinib may exert an antiarrhythmic effect by reducing Ryr2 hyperphosphorylation and ameliorating sarcoplasmatic calcium handling, thus reducing atrial susceptibility to non-re-entrant tachyarrhythmias. Nevertheless, it should be noted that the significant improvement in respiratory function experienced by our patient may have played a role in reducing the risk of relapse of MAT.

Further large prospective studies are needed to assess the role of highly selective MEK inhibitors in RASopathies.

## Figures and Tables

**Figure 1 genes-13-01503-f001:**
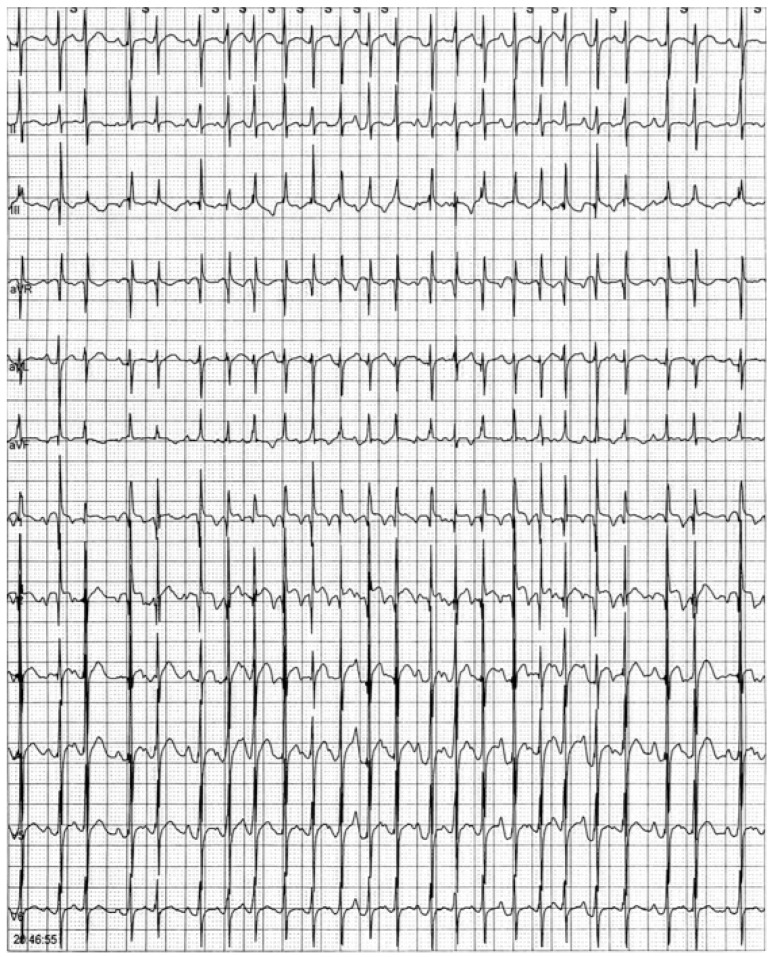
ECG Holter monitoring showing an episode of multifocal atrial tachycardia.

**Figure 2 genes-13-01503-f002:**
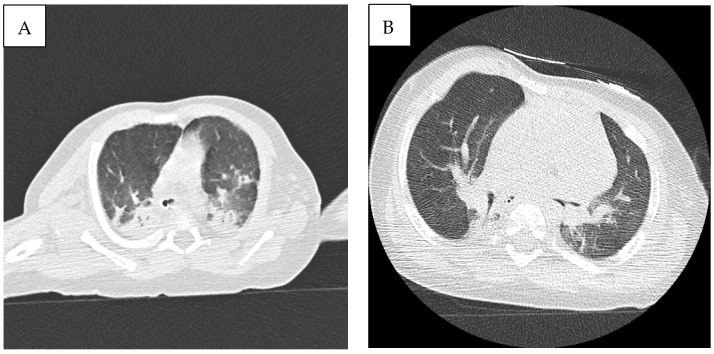
(**A**) High-resolution CT scan (HRCT) showing bilateral parenchymal lung consolidation, involving both superior and inferior lobes, mainly with posterior extension, associated with diffuse *ground glass* opacities, consistent with pulmonary lymphangiectasia/acute respiratory distress. (**B**) Follow-up chest CT showing a significant reduction in lung consolidation areas.

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
