# Peer review of "Severe Lymphatic Disorder and Multifocal Atrial Tachycardia Treated with Trametinib in a Patient with Noonan Syndrome and SOS1 Mutation"

_genes, 2022, doi:10.3390/genes13091503_

Round 1
Reviewer 1 Report
Summary: Lioncino et al describe the case of a preterm neonate with SOS1 related Noonan syndrome who developed perinatal respiratory failure from presumed pulmonary lymphangiectasia and multifocal atrial tachycardia. They describe how use of targeted MEK inhibition with trametinib led to improvement of pulmonary pathology and atrial arrhythmia. This is an important addition to the emerging and growing body of literature of molecularly targeted therapy for rasopathies. This report will further strengthen the clinical evidence for use of these therapies and ideally provide a basis for more definitive clinical trials.
General comments:
Overall, the authors present a strong case that initiation of trametinib was temporally associated in improvement in multiple Ras/MAPK pathway pathologies. However, some clinical data was not presented that could further strengthen the authors’ arguments of trametinib mediated effect and possible mechanisms of action.
Specific Comments:
1. Please describe the fetal morphologic abnormalities in relation to whether the patient would meet criteria for non-immune hydrops. I.e. were pleural effusions present in addition the polyhydramnios. This could indirectly support that the pulmonary pathology discussed in the case is related to lymphatic dysplasia given the association with non-immune fetal hydrops
2. Similarly, please discuss whether the pleural effusions seen were suggestive of a chylous versus a serous effusion and support this if fluid studies were obtained from the perinatal thoracentesis. As direct lymphatic imaging was not available, secondary evidence that the effusions were related to lymphatic dysplasia would be helpful
3. Please discuss whether there was any evidence of development of hypertrophic cardiomyopathy during initial or serial monitoring. In workgroup discussion and my own practice, trametinib initiation in the setting of cardiomyopathy is associated with a quick improvement in markers of diastolic dysfunction (i.e. BNP) on the order of days. If pleural effusions and echocardiographic evidence are suggestive of more significant diastolic dysfunction resulting in pulmonary edema (either from hypertrophic cardiomyopathy or arrhythmia related ventricular dysfunction), this would be an important alternative hypothesis to consider for improvement in pulmonary pathology with initiation of trametinib as opposed to lymphatic mediated effects.
4. Please also discuss the timeline of improvement of the multifocal atrial tachycardia in relation to trametinib initiation. If the MAT resolved/improved with the trametinib reaching steady state serum levels (~48-72 hours), this would support a direct MEK mediated arrhythmia signaling pathway as opposed to secondary to respiratory or other improvements in the patient’s clinical state.
5. Figure 1 is not necessary as a de novo pathogenic variant is adequate for description.
6. Recommend combining figures 3 and 4 to allow for side by side comparison of pulmonary disease with clear visualization of timeline between imaging. Ideally the imaging plane would be matched if possible.
7. Supplemental figure 1: Please define the exam timepoints, I am assuming the units are most after start of treatment. Please also define the studies included in “biochemical/full cell count” (could be as a foot note).
Reviewer 2 Report
In the Case Report titled ”Severe lymphatic disorder and multifocal atrial tachycardia treated with trametinib in a patients with Noonan syndrome and SOS1 mutation” the authors report a critical case of a pre-term new-born who was admitted for the occurrence of severe respiratory distress and subentrant MAT and was treated with trametinib.
General comments:
The manuscript reports an important finding where the newborn was treated with trametinib following a previous report by Andelfinger et al. cited by the author. The case is very well presented. The manuscript is well written and easy to follow except for the brief introduction.
I just have two minor comments :
- In the title, the authors should correct " in a patients" to "in a patient"
- The authors have provided a very brief introduction. The authors should provide detailed information on the disease, different majorly involved genes, mutations causing the disease and the current treatment.
Round 2
Reviewer 1 Report
The authors have thoroughly addressed this reviewer's prior critiques and comments.